# MiR-24-3p Conservatively Regulates Muscle Cell Proliferation and Apoptosis by Targeting Common Gene *CAMK2B* in Rat and Cattle

**DOI:** 10.3390/ani12040505

**Published:** 2022-02-17

**Authors:** Ge Yang, Mingli Wu, Xinqi Liu, Fuwen Wang, Mei Li, Xiaoya An, Fuxia Bai, Chuzhao Lei, Ruihua Dang

**Affiliations:** Key Laboratory of Animal Genetics, Breeding and Reproduction of Shaanxi Province, College of Animal Science and Technology, Northwest A&F University, Yangling, Xianyang 712100, China; geyang0125@163.com (G.Y.); wumingli1993@163.com (M.W.); novel548@163.com (X.L.); wangfuwen2016@nwafu.edu.cn (F.W.); 18702262807@163.com (M.L.); axy19591558024@163.com (X.A.); 18821700428@163.com (F.B.); leichuzhao1118@126.com (C.L.)

**Keywords:** miR-24-3p, muscle development, C2C12 cell, CCK-8, double luciferase test, *CAMK2B*

## Abstract

**Simple Summary:**

MicroRNAs (miRNAs) play important roles in the development of skeletal muscle. Here, a comparative analysis of four skeletal muscle transcriptomes of cattle, rat, goat, and pig showed that miR-24-3p may conservatively regulate muscle development. There is a large proportion of the target genes of miR-24-3p shared by cattle, rat, goat, and pig. GO (Gene ontology) and KEGG (A Kyoto Encyclopedia of Genes and Genomes) enrichment analysis showed that these genes are enriched in multiple cell functions and signal pathways that are closely related to muscle development. In rat and cattle, a double luciferase test showed that three shared target genes, *WNT4 (Wnt Family Member 4)*, *CAMK2B (Calcium/Calmodulin Dependent Protein Kinase II Beta)*, *TCF7 (Transcription Factor 7)* were targeted by mmu-miR-24-3p and bta-miR-24-3p. The three shared target genes (*WNT4*, *CAMK2B*, and *TCF7*) were all targets in the Wnt signaling pathway, which plays an important role in muscle proliferation in rat and cattle. The shared target gene (*CAMK2B*) rat was increased significantly after the inhibition of miR-24-3p in rat and cattle. This study will be a good foundation by which to improve our understanding of the functions of miRNAs in the regulation of muscle development.

**Abstract:**

Skeletal muscle plays an important role in the growth and development of meat animals. MicroRNAs (miRNAs) can participate in the regulation of muscle development-related functions; however, there have been few reports on whether there are related miRNAs that conservatively regulate muscle development among different species. In this study, the miRNA transcriptome sequencing data of the muscle tissue of cattle, rat, goat, and pig showed that miR-24-3p may conservatively regulate muscle development in these species. Furthermore, mmu-miR-24-3p can positively regulate C2C12 cell proliferation and apoptosis by regulating key proliferation and apoptosis genes in muscle development, which was verified by CCK-8 and RT-qPCR. Bta-miR-24-3p can also positively regulate the proliferation and apoptosis of bovine muscle primary cells by regulating key proliferation and apoptosis genes in the process of muscle development, as verified by CCK-8 and RT-qPCR. The target genes of miR-24-3p in cattle, rat, goat, and pig, which include a large proportion of target genes shared among the four species, are enriched in multiple cell functions and signal pathways that are closely related to muscle development, as revealed by GO and KEGG enrichment analysis. A double luciferase test showed that the shared target genes *WNT4*, *CAMK2B*, and *TCF7* were targeted by mmu-miR-24-3p in rat and bta-miR-24-3p in cattle. These three shared target genes *WNT4*, *CAMK2B*, and *TCF7* are involved in the Wnt signaling pathway, which showed that miR-24-3p plays an important role in rat and cattle. The shared target gene (*CAMK2B*) in rat and cattle increased significantly after the inhibition of miR-24-3p by RT-qPCR. The findings of this study contribute to a better understanding of the role of miR-24-3p in the regulation of muscle development.

## 1. Introduction

MicroRNAs (miRNAs) are non-coding RNAs that can regulate gene expression at the post-transcriptional level. Since the discovery of Lin-4 miRNA in *Caenorhabditis elegans*, an increasing number of miRNAs and their targets have been identified [1]. Their classic mode of action is to regulate related biological functions through targeted binding to the 3′-UTR region of mRNA [2]. By regulating the expression of multiple target genes in organisms, miRNA induces changes in a variety of biological processes, including cell apoptosis, proliferation, and cell senescence [3]. MiRNAs play important roles in many important biological processes including skeletal muscle myogenesis [4]. The knockout of Dicer, specifically in muscles, can cause muscle hypoplasia and early death in neonatal mice, which further illustrates that miRNA plays an important role in muscle development [5]. Early findings indicate that gga-miR-200a can inhibit the expression level of its target gene *GRB2* (*growth factor receptor-bound protein 2*) and may participate in the cell differentiation and proliferation of chicken breast muscle by combining with the 3′UTR of *GRB2* [6]. miR-143 can regulate the proliferation and differentiation of bovine skeletal muscle satellite cells by changing the expression of *IGFBP5* (*insulin-like growth factor binding protein 5*) [7]. The study also found that the levels of miR-1 and miR-206 increased significantly during the differentiation of myoblasts [8]. MiR-24-3p, as a kind of miRNA, mainly plays an important role in cancer. Studies have found that miR-24-3p can promote lung cancer cell migration and proliferation by targeting *SOX7* (*SRY-box transcription factor 7*) [9].

Skeletal muscle plays an indispensable role in the body as it is dynamic tissue of the human body and accounts for 40% of the body’s total weight [10]. In addition, the study found that myogenesis requires post-transcriptional regulation by myogenic microRNAs (myomiRNAs) [11]. However, the mechanism behind skeletal muscle cell proliferation and apoptosis is complex, and its regulation process is still unclear. Hence, it is important to clarify the molecular mechanism of skeletal muscle development. Several myo-miRNAs are upregulated by, e.g., MyoD and myogenin, during myogenic differentiation including miR-1 [12], miR-133 [13,14], and miR-206 [15], each of which inhibits various target RNAs [16].

Previous studies have found many miRNAs that can participate in the regulation of muscle development-related functions [17,18], but there are few reports on whether there are related miRNAs that regulate muscle development conservatively among species. Therefore, based on the results of the high-throughput sequencing of muscle tissue miRNAs of different species (cattle, rat, goat, and pig), this study found that miR-24-3p may be a potential miRNA that can conservatively regulate muscle development. Through a series of experimental verifications, it was shown that miRNA-24-3p can promote proliferation and inhibit apoptosis of C2C12 cells and bovine muscle cells. In order to further explore its conservative regulation mechanism, the dual luciferase reporter system was used to verify the targeting relationship between miR-24-3p and the shared target between species, and it was further speculated that miR-24-3p might regulate shared target gene groups widely among species to affect muscle development.

## 2. Materials and Methods

### 2.1. Ethics Statement

All the experimental designs and procedures were approved by the Regulations for the Administration of Affairs Concerning Experimental Animals (Ministry of Science and Technology, China, 2004). The study was approved by the Institutional Animal Care and Use Committee of Northwest A&F University (approval number: 20171208–010, 8 December 2017).

### 2.2. Identification of Conserved miRNAs of Cattle, Rat, Goat, and Pig

The miRNA was identified from muscle transcriptome data; these data were downloaded from the National Center for Biotechnology (NCBI) database (http://asia.ensembl.org/info/data/ftp/index.html accessed on 20 December 2021), including the muscle tissues of cattle [19], pig [20], goat [21], and rat [22]. Based on these sequences, we used Bioinformatics and Evolutionary Genomics software (http://bioinformatics.psb.ugent.be/cgi-bin/liste/Venn/calculate_venn.htpl accessed on 25 December 2021) to perform a Veen analysis to screen and obtain all the miRNAs that are stably expressed [23]. After obtaining the miR-24 sequence information from the miRBASE database [24,25], the sequence was then compared using MEGA5.0 software to construct a phylogenetic tree [26] (Appendix A).

### 2.3. Collection of Cell Samples (C2C12, 239T, and Qinchuan Cattle Fetus)

The C2C12 cell line (accession number: CRL-1772) and the 293T cell line (accession number: CRL-3216) were obtained from Procell Life Science & Technology Co., Ltd. [27,28]. The 293T cell line (accession number: NM_131629) was obtained from Procell Life Science & Technology Co., Ltd. (Procell Life Science & Technology Co., Ltd., Beijing, China) [28]. The Qinchuan cattle fetus in third month of gestation samples were collected from a beef cattle slaughterhouse near Xi’an. The tissue spectrum included the heart, liver, spleen, lung, kidney, muscle, and small intestine. The bovine muscle cell culture samples were aseptically processed for cell culture use. The study was approved by the Biomedical Ethics Committee of Northwest A&F University.

### 2.4. Cell Culture and RNA Isolation and Reverse Transcription-Quantitative Polymerase Chain Reaction (RT-qPCR)

The C2C12 cells were procured from Procell Life Science & Technology Co., Ltd. The 293T cells were procured from Procell Life Science & Technology Co., Ltd. (Procell Life Science & Technology Co., Ltd., Beijing, China). These C2C12 cells, the bovine muscle cells, and the 293T cells were maintained at 37 °C in a 5% CO_2_ humidified incubator and were grown in Dulbecco’s Modified Eagle’s Medium (DEME) supplemented with 10% fetal bovine serum (FBS) [29]. When the confluence of cells reached 80–90%, they were digested with 0.25% trypsin and passaged to a new culture dish at a ratio of 1:3. The cells were transfected with 50 nM of control or mimic for miR-24b-3p mixed with Opti-MEM and Lipofectamine RNAiMAX (Invitrogen, Bao Bioengineering (Dalian) Co., Ltd., Dalian, China) according to manufacturer’s protocol [30]. All the analyses were performed in triplicate. The cells were lysed, and total RNA was extracted using TRIzol reagent (Takara, Bao Bioengineering (Dalian) Co., Ltd., Dalian, China) according to the manufacturer’s protocol and transcribed into cDNA using a Reverse Transcription Kit (Takara, Bao Bioengineering (Dalian) Co., Ltd., Dalian, China) [31]. The expression patterns of the target genes and the transcriptional responses of the target genes to the muscle were investigated using reverse transcription-quantitative polymerase chain reaction (RT-qPCR). According to the manufacturer’s instructions, after DNase treatment, 1000 ng of total RNA was reverse-transcribed to single-strand cDNA using a HiScript^®^ III 1st Strand cDNA Synthesis Kit (+gDNA wiper)(Nanjing novozan Biotechnology Co., Ltd., Nanjing, China). The primer pairs of the target genes were used (Appendix A). Before RT-qPCR analysis, the standard curves for the primer pair of the target genes were generated by regression of the Cq values and a series of ten-fold cDNA dilutions. Primer amplification efficiency was calculated from the slope of the corresponding standard curve, and the efficiency of the target genes. The hypoxic-stable reference gene β-actin was used as the control (Appendix A). RT-qPCR was performed using the ChamQTM Universal SYBR^®^ qPCR Master Mix (Nanjing novozan Biotechnology Co., Ltd., Nanjing, China) with the following thermal cycling conditions: 95 °C for 30 s, 40 cycles of 95 °C for 10 s, and 60 °C for 30 s. Each experiment was performed independently three times. The relative expression levels of the target genes were normalized to that of β-actin quantification using the 2^−^^△△Ct^ method [31].

### 2.5. Validation of miR-24-3p Transfection Efficiency in C2C12 Cells and Primary Myoblasts Cells and Cell Counting Kit-8 (CCK-8) Cell Viability Assay

The two groups (miR-24-3p mimics and mimics–negative control, and inhibitor and inhibitor–negative control) were transfected into C2C12 cells and primary myoblasts cells, which were in the logarithmic phase. After 24 h, total RNA was extracted and specifically reversed, and the overexpression efficiency and inhibition efficiency of the miRNA were detected by RT-qPCR as above (Appendix A). The C2C12 cells and primary myoblast cells transfected with miR-24-3p mimics and mimics–negative control (mimics-NC) were seeded in 96-well plates at a density of 5 × 10^3^ per well and cultured for 0 days, 1 day, 2 days, and 3 days. We used 100 µL medium per well. Cell viability was assessed using the Cell Counting Kit-8 (CCK-8, Dojindo, Shanghai, China) at days 0, 1, 2, and 3 [32].

### 2.6. Target Gene Prediction and Functional Enrichment Analysis

The miR-24-3p target genes of different species were predicted using TARGETSCAN (http://www.targetscan.org/vert_71/ accessed on 30 December 2021) software [33]. To analyze the main function of the differentially expressed genes or miR-24-3p target genes, gene ontology (GO) enrichment analysis was performed using the GOseq R package [34]. A Kyoto Encyclopedia of Genes and Genomes (KEGG) enrichment analysis of the predicted target genes was performed using KOBAS software [35]. GO terms and KEGG pathways with *p* < 0.05 were considered significantly enriched.

### 2.7. Plasmid Construction and Recombinant Vector Cell Transfection

The target sequences were predicted using NCBI database, and the primers of these target genes *MEIS2* (*Meis Homeobox 2*), *MAPK7* (*Mitogen-Activated Protein Kinase 7*), *UCP2* (*Uncoupling Protein 2*), *SNX1* (*Sorting Nexin 1*), *CPS1* (*Carbamoyl-Phosphate Synthase 1*), *HACD3* (*3-Hydroxyacyl-CoA Dehydratase 3*), *WNT4* (*Wnt Family Member 4*), *CAMK2B* (*Calcium/Calmodulin Dependent Protein Kinase II Beta*), *TCF7* (*Transcription Factor 7*) were designed using Primer 5. The primer information of mmu-miR-24-3p and bta-miR-24-3p were listed in Appendix A. The genomic DNA mRNA 3′-UTR sequence of these target genes was cloned from genomic DNA using polymerase chain reaction (PCR) amplification. When we obtained the correct DNA products, we inserted these target genes’ genomic DNA into the psiCHECK2 vector (Promega, Madison, WI, USA). The recombinant vectors were extracted using an Endofree Mini Plasmid Kit II (Tiangen, Beijing, China) extraction kit, followed by confirmation with enzyme digestion sequencing. After sequencing verification, the successfully constructed vectors were used for cell transfection. The transfection procedure was performed following the manufacturer’s instructions for the transfection reagent (Thermo Fisher Scientific, Waltham, MA, USA).

### 2.8. Dual-Luciferase Reporter System Assay

The C2C12 cell lines were co-transfected with the constructed psiCHECK2-mmu-miR-24-3p vectors, and the bovine primary myoblasts cell lines were co-transfected with the constructed bta-miR-24-3p vectors. We performed the dual-luciferase vector following the manufacturer’s instructions regarding the Thermo transfection reagent. To test the effect of the miRNA on its potential target gene’s luciferase activity, we referred to the detailed procedures of the luciferase reporter assay kit instructions (Promega, Madison, WI, USA), and the Renilla luciferase signal was normalized to the firefly luciferase signal.

### 2.9. Statistical Analysis

The mean difference between groups was calculated by one-way ANOVA, and the mean difference between the two groups was tested by an independent samples *t* test. The test data were analyzed using one-way ANOVA with SPSS 23.0 (SPSS Inc., Chicago, IL, USA) and were expressed as mean ± SE [36]. The level of statistical significance was set at *p* < 0.05 for all the analyses. All the experiments were performed independently three times (Appendix A).

## 3. Results

### 3.1. MiR-24-3p Is Conserved in Different Animals (Cattle, Rat, Goat, and Pig)

We identified a total of 54, 32, 41, and 48 miRNAs of cattle, pig, goat, and rat from the NCBI database, respectively. The Venn diagram of these miRNAs showed that there were three miRNAs (miR-199a-3p, miR-24-3p, and miR-30a-5p) that were shared among the four species (Figure 1A,B). The NJ tree of the miR-24 sequence of different species showed that miR-24-1 and miR-24-2 divided into two groups according to species divergence (Figure 1B). We also found that the mature sequence of cattle miR-24-3p is similar to that of goat, chickens, humans, and rat, and it presented a high degree of evolutionary conservation in domestic animals (Figure 1C). These results showed that miR-24-3p may be an epigenetic regulatory molecule that is not restricted to species and has a broad regulatory role in muscle development (Figure 1, Appendix A).

### 3.2. MiR-24-3p Promotes C2C12 Cell Proliferation and Inhibits Their Apoptosis

To investigate the biological role of miR-24-3p in C2C12 cell proliferation and apoptosis, we transfected C2C12 cells with miR-24-3p mimics, mimics–negative control (mimics-NC), inhibitor, and inhibitor–negative control (inhibitor-NC), then we detected the expression of the proliferation and apoptosis marker gene. Compared with the NC group, the expression level of miR-24-3p was significantly increased (Figure 2A). We also found that the overexpression efficiency of miR-24-3p was almost equal to the inhibition efficiency, both being 20-fold increased (Figure 2A,B). Using the CCK-8 regent to detect the proliferative state of bovine muscle cells, the OD value of the miR-24-3p group was significantly increased (Figure 2C). Furthermore, the mRNA levels of the proliferation and apoptosis-relative apoptosis marker genes (*CASPASE9*, *CASPASE3*, and *CASPASE8*) were significantly decreased after miR-24-3p was overexpressed (Figure 2D–F), and the expressions of the proliferation-related genes (the *CYCLINE*, *PCNA*, and *BAX* genes) were significantly increased (Figure 2G–I). After inhibiting the expression of miR-24-3p in C2C12 cells, the expression levels of the proliferation-related genes (*CASPASE9*, *CASPASE3*, *CASPASE8*, and *CYCLIND*) were significantly increased (Figure 2D–F,J), and the expression levels of apoptosis marker genes (*PCNA)* were significantly decreased (Figure 2I). All these results indicated that miR-24a-3p could promote C2C12 cell proliferation and inhibit its apoptosis (Figure 2, Appendix A).

### 3.3. MiR-24-3p Promotes Bovine Muscle Cell Proliferation and Inhibits Their Apoptosis

In order to study the regulatory effect of miR-24-3p on cattle muscle development, the expression profile of miR-24-3p in seven tissues related to cattle muscle development, including fetal and adult bovine heart, liver, spleen, lung, kidney, muscle, and small intestine, were investigated using RT-qPCR. These results showed that the expression level of miR-24-3p in various tissues of adult cattle was significantly higher than that of fetal cattle (Figure 3A). The results of the RT-qPCR of miRNA-24-3P in seven tissues (heart, liver, spleen, lung, kidney, muscle, and small intestine) showed that miR-24-3p was highly expressed in muscles and the heart, with these two tissues belonging to skeletal muscle and cardiac muscle, respectively (Figure 3A). The results showed that the expression of miR-24-3p after the transfection of mimics was about 60 times greater than that of the NC group (Figure 3B), and the expression of miR-24-3p after transfection of inhibitor was 1/50 lower that of the NC group (Figure 3C). By using the CCK-8 regent to detect the proliferative state of bovine muscle cells, the OD value of the miR-24-3p group was significantly increased (Figure 3D). To determine the role of miR-24-3p in cattle muscle cell apoptosis and proliferation, we detected the mRNA expression level of some relative genes. The results showed that the expression of the proliferation-related genes (*BCL2* and *CDK2)* was significantly increased after miR-24-3p was overexpressed, while the expression of the proliferation-related gene (*CASPASE9*) was significantly decreased after miR-24-3p was inhibited (Figure 3E). Together, the results elucidate that miR-24-3p could promote cattle muscle cell proliferation and inhibits their apoptosis (Figure 3, Appendix A).

### 3.4. Prediction of miR-24-3p Target Genes and Annotation in Different Species

There were 315 target genes of miR-24-3p shared among the four species (cattle, mice, humans, and rat), accounting for 46.46% of the total number of miR-24-3p target genes according to the Venn analysis. This may be an important basis by which miR-24-3p conservatively regulates muscle development in different species (Figure 4A, Table 1). There were 417 target genes shared between the two species (rat and cattle), and these common target genes account for 78.7% of all the target genes in rat and 61.5% of all the target genes in cattle (Figure 4B, Table 1). The large proportion of the same target genes may be related to the previously demonstrated ability of miR-24-3p to regulate the proliferation and apoptosis of C2C12 cell lines and bovine primary muscle cells (Figure 4, Table 1).

The GO analysis of the target genes of miR-24-3p shared by rat and cattle showed that they were enriched in the negative regulation of cell proliferation, protein autophosphorylation, protein tyrosine phosphatase activity, the cytoplasmic mRNA processing body, and neural crest cell migration (Figure 4C). The KEGG analysis of the common target genes of mmu-miR-24-3p and bta-miR-24-3p found that the target genes were predicted to be enriched in the MAPK signaling pathway, the cAMP signaling pathway, Axon guidance, the Wnt signaling pathway, and basal cell carcinoma (Table 1). These combined results showed that the shared target genes of mmu-miR-24-3p and bta-miR-24-3p are enriched in multiple cell functions and signal pathways that are closely related to muscle development (Figure 4C, Table 1). Therefore, it is speculated that the role of miR-24-3p in regulating muscle development in different species may be achieved by regulating the shared target genes (Figure 4).

### 3.5. The Specific and Common Target Genes’ Expression Profile of miR-24-3p in Cattle and Rat

The double luciferase test was performed for miR-24-3p and their predicted target genes using 293T cells that were transfected with miR-24-3p mimics/mimics NC/inhibitor/inhibitor-NC (Figure 5A,C,E). The unique target genes *MEIS2*, *MAPK7*, and *UCP2* in rat were targeted by mmu-miR-24-3p (Figure 5A). The unique target genes *SNX1*, *HACD3*, and *CPS1* in cattle were targeted by bta-miR-24-3p (Figure 5C; Appendix A). The results showed that the shared target genes *WNT4*, *CAMK2B*, and *TCF7* were targeted by mmu-miR-24-3p and bta-miR-24-3p (Figure 5E). For cattle, after inhibition of bta-miR-24-3p, the expression of the *SNX1* and *HACD3* genes increased significantly (Figure 5D). Of these three common target genes (*WNT4*, *CAMK2B*, and *TCF7*), we found that after overexpression of miR-24-3p, the expression of the target gene *CAMK2B* increased significantly (Figure 5F, Appendix A).

## 4. Discussion

miR-24 is highly conserved among various species. Studies have shown that miR-24 can promote human adipogenesis and differentiation [37,38]. miR-1 and miR-206, with similar sequences, are involved in the promotion of myogenic differentiation [8,39]. miR-24-3p can regulate human and rat skeletal muscle [40,41]. Previous studies have found that microRNA can bind to the 3′-UTR region of the target gene to perform its function through a classical approach, but the 3′-UTR region of genes in different species is relatively not conservative [2]. We speculate that the sequences of some microRNAs themselves should be conservative. In order to verify our conjecture, we identified microRNAs from the miRNA sequencing results of muscle tissues of four species of cattle, pig, goat, and rat using NCBI. The Venn analysis showed that there were only three microRNAs shared by these four species, namely, miR-199a-3p, miR-24-3p. and miR-30a-5p according to the miRNA sequencing results of the muscle tissues of four species of cattle, pig, goat, and rat using NCBI (Figure 1A). It is preliminarily believed that the function of miR-24-3p in regulating muscle development is conservative in these species. miR-24-3p can be formed by the splicing of two precursor sequences of miR-24-1 and miR-24-2 [42,43]. The phylogenetic tree of the precursor sequences of miR-24-3p from different species showed that bta-miR-24-2 is close to mmu-miR-24-2, verified by sequence alignment (Figure 1). It is further speculated that miR-24-3p may be an epigenetic regulatory molecule that is not restricted to these species and has a wide-ranging regulatory role in muscle development [44].

Previous studies generally believed that miR-24-3p could inhibit the proliferation and migration of cancer cells [45]. However, there are relatively few studies on the involvement of miR-24-3p in skeletal muscle. Some researchers have proved that miR-24 can protect the myocardium by promoting cardiomyocyte proliferation and reducing myocardial fibrosis [46]. It was also found that miR-24-3p can promote the proliferation and differentiation of human skeletal muscle by regulating the TGF-β/Smad pathway [47,48], and that it can inhibit the apoptosis of smooth muscle cells and reduce skeletal muscle fibrosis [49]. It was also found that the loss of microRNA-23-27-24 clusters in mouse skeletal muscle had no effect on skeletal muscle development and exercise-induced muscle adaptation [50]. In order to verify the function of miR-24-3p in rat and cattle muscle development, this experiment used C2C12 cells and bovine muscle cells to verify the effect of miR-24-3p on rat and cattle muscle cell proliferation and apoptosis. We detected the proliferation-related genes’ expression level, such as *CYCLINE* [51], *PCNA* [52], *BAX* [53], *CASPASE9*, *CASPASE3*, and *CASPASE8* [54] in rat, *BCL2* [55], *CDK2* [56], and *CASPASE9* in cattle [54]. In rat, the C2C12 cell number was significantly increased with miR-24-3p mimics compared to the mimics–negative control (mimics-NC) group. After miR-24-3p was overexpressed, the expression level of three proliferation-related genes *CYCLINE*, *PCNA*, and *BAX* were significantly increased; however, the expression level of three apoptosis-related genes *CASPASE9*, *CASPASE3*, and *CASPASE8* were significantly decreased. These three results show that miR-24-3p could significantly promote C2C12 cell proliferation and inhibit its apoptosis [57]. In cattle, the RT-qPCR results for miR-24-3p in seven different tissues showed that miR-24-3p was highly expressed in muscle (skeletal muscle) and the heart (cardiac muscle) compared to the other five tissues, indicating that miR-24-3p plays an important role in muscle development. Furthermore, the cell number of bovine muscle cells in the miR-24-3p group were significantly increased compared to the NC group. The expression level of proliferation-related genes (*BCL2* and *CDK2*) were significantly increased after miR-24-3p was overexpressed, while the expression of proliferation key gene (*CASPASE9*) genes was significantly decreased after miR-24-3p was inhibited. These four results showed that miR-24-3p can promote bovine muscle cell proliferation and inhibits their apoptosis [57]. This study found, for the first time, that miR-24-3p also has functions related to the regulation of muscle cell development in bovine muscle cells. This is consistent with the existing reports on the regulatory effect of miR-24-3p on muscle development in mice and humans (Figure 2 and Figure 3) [57].

The GO and KEGG analysis of the target genes shared by mmu-miR-24-3p and bta-miR-24-3p showed enrichment of important cell functions and signal pathways related to muscle development, including negative regulation of cell proliferation, protein autophosphorylation, the MAPK signaling pathway, the cAMP signaling pathway, Axon guidance, the Wnt signaling pathway, and basal cell carcinoma. These results show that the target genes shared by miR-24-3p in rat and cattle are enriched in multiple cell functions and signal pathways that are closely related to muscle development (Figure 4). The shared target genes *WNT4*, *CAMK2B*, and *TCF7* were both targeted by mmu-miR-24-3p in rat and bta-miR-24-3p in cattle, which was proved by the double luciferase test performed for miR-24-3p and their predicted target genes using 293T cells that were transfected with miR-24-3p mimics/mimics NC/inhibitor/inhibitor-NC. Moreover, the three-shared target genes (*WNT4*, *CAMK2B*, and *TCF7*) were all targets in the Wnt signaling pathway, which was consistent with the previous studies claimed that Wnt signaling pathway play an important role in muscle proliferation in rat and cattle [58,59]. After overexpression of miR-24-3p, the expression of the shared target gene *CAMK2B* increased significantly in rat and cattle. Previous studies showed that *CAMK2B* plays an important role in muscle development [60,61]. These results show that *CAMK2B* is a conserved target gene of miR-24-3p in cattle and rat that plays an important role in muscle development in cattle and rat, which is consistent with the results of previous studies (Figure 5) [60,61].

## 5. Conclusions

In summary, with the miRNA transcriptome sequencing data of muscle tissue in cattle, rat, goat, and pig, we found that miR-24-3p plays a potential role in regulating muscle development in animals. CCK-8 and RT-qPCR analysis showed that mmu-miR-24-3p can positively regulate C2C12 cell proliferation and apoptosis, and bta-miR-24-3p can also positively regulate the proliferation and apoptosis of bovine muscle primary cells. The GO and KEGG enrichment analysis results showed that the target genes of miR-24-3p in cattle, rat, goat, and pig are closely related to muscle development. The shared target genes *WNT4*, *CAMK2B,* and *TCF7* were both targeted by mmu-miR-24-3p and bta-miR-24-3p using the double luciferase test in rat and cattle. These three shared target genes, *WNT4*, *CAMK2B*, and *TCF7*, are involved in the Wnt signaling pathway, which showed that miR-24-3p plays an important role in rat and cattle. After inhibition of miR-24-3p, the target gene *CAMK2B*, which plays an important role in muscle development, increased significantly, indicating that miR-24-3p is a conservative miRNA and that it can regulate the most predicted target gene (*CAMK2B*) to influence the development of muscle.

## Figures and Tables

**Figure 1 animals-12-00505-f001:**
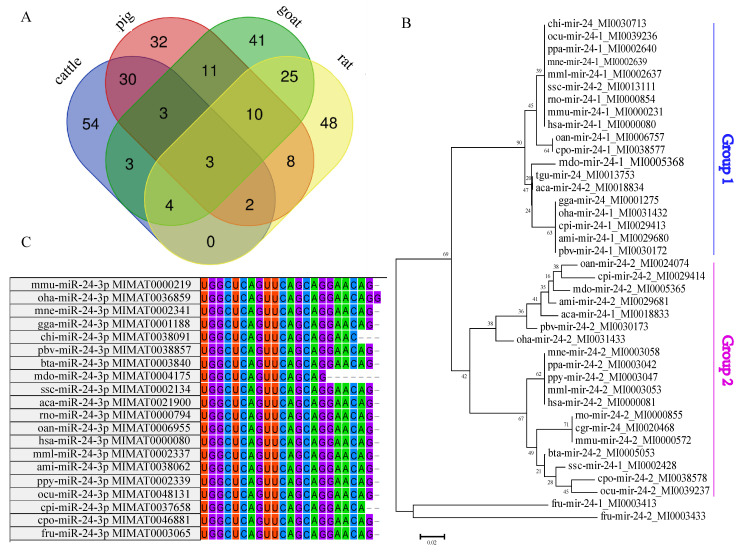
Identification of common miRNA in the muscle tissue of different species. (**A**) The Venn analysis of the top 100 miRNAs with high expression in the muscle tissues of cattle, pig, goat, and rat. (**B**) The phylogenetic tree of miR-24 in different species. (**C**) Multiple sequence alignment of miR-24-3p sequences in different species.

**Figure 2 animals-12-00505-f002:**
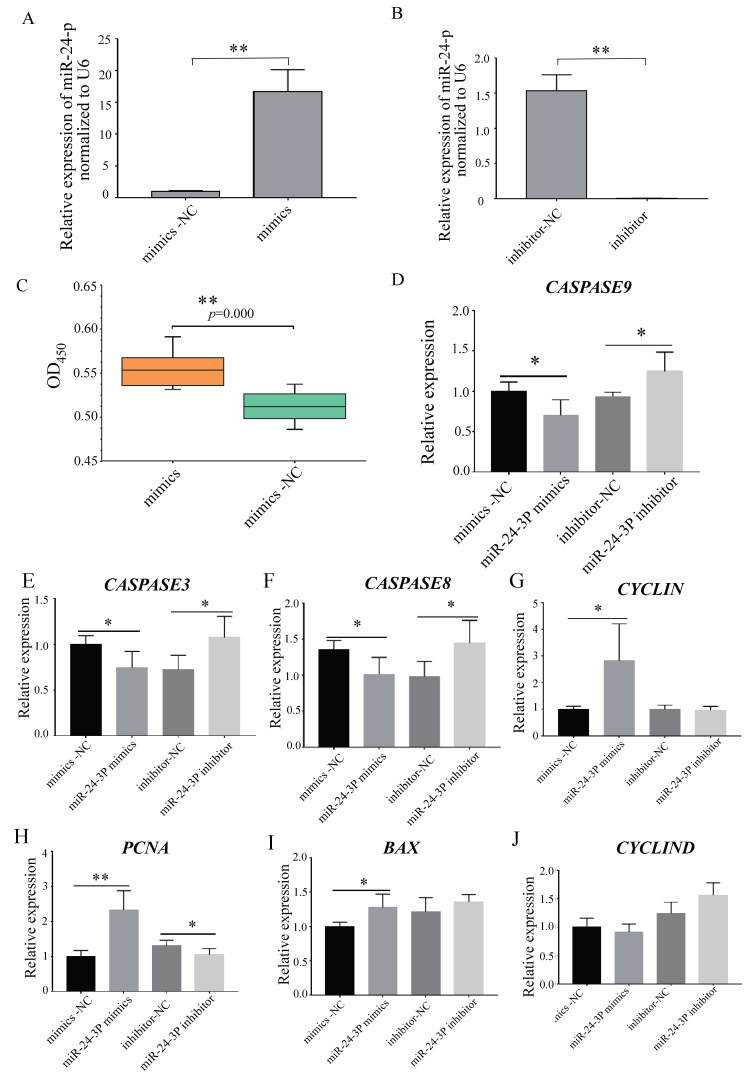
MiR-24-3p regulates C2C12 cell proliferation and apoptosis. (**A**) Overexpression efficiency detection of mmu-miR-24-3p by RT-qPCR. (**B**) Inhibition efficiency detection of miR-24-3p by RT-qPCR. (**C**) Cell proliferation status was detected at 450 nm with the CCK-8 reagent after an increase in miR-24-3p. (**D**–**J**) The expression of the *CASPASE9*, *CASPASE3*, *CASPASE8*, *CYCLINE*, *CYCLIND*, *PCNA*, and *BAX* genes after overexpression and inhibition of miR-24-3p, respectively. * *p* < 0.05, ** *p* < 0.01.

**Figure 3 animals-12-00505-f003:**
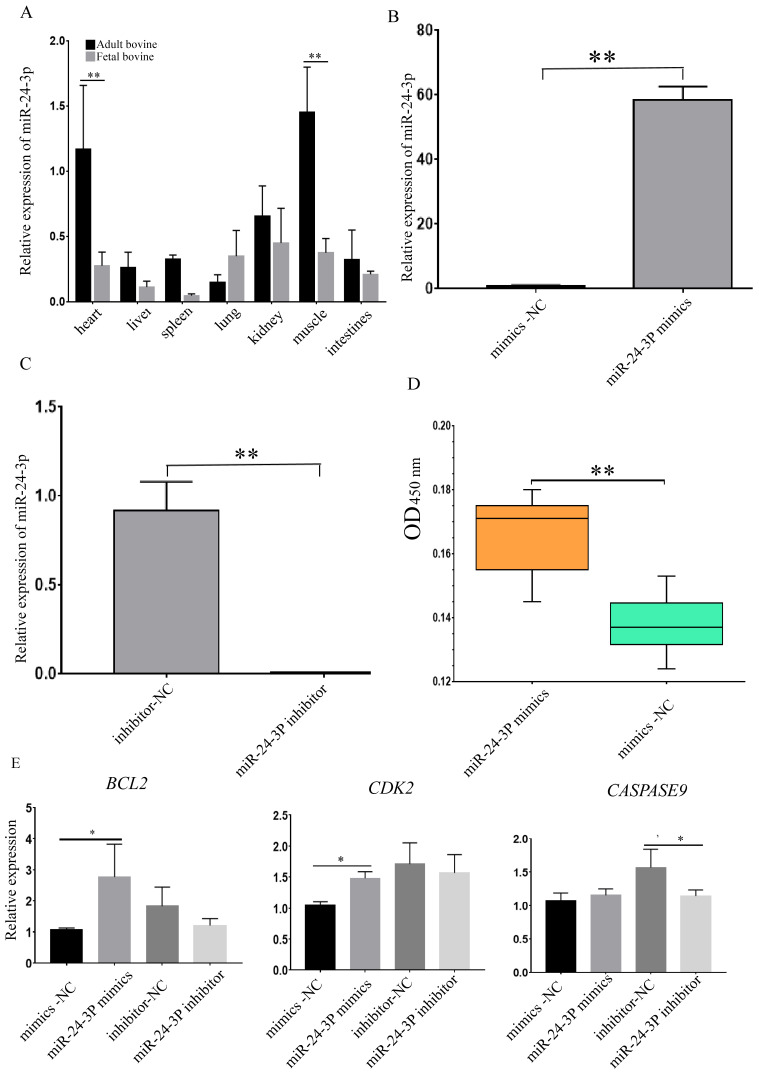
MiR-24-3p regulates bovine muscle cell proliferation and apoptosis. (**A**) RT-qPCR analysis of miR-24-3p in different tissues of fetal and adult cattle. (**B**) Overexpression efficiency detection of miR-24-3p in cattle muscle cells by RT-qPCR. (**C**) Inhibition efficiency detection of miR-24-3p in cattle muscle cells by RT-qPCR. (**D**) Cell proliferation status was detected at 450 nm wavelength with CCK-8 reagent after an increase in miR-24-3p. (**E**) The expression of *CDK2*, *BCL2*, and *CASPASE9* in cattle muscle cells by RT-qPCR, respectively. * *p* < 0.05, ** *p* < 0.01.

**Figure 4 animals-12-00505-f004:**
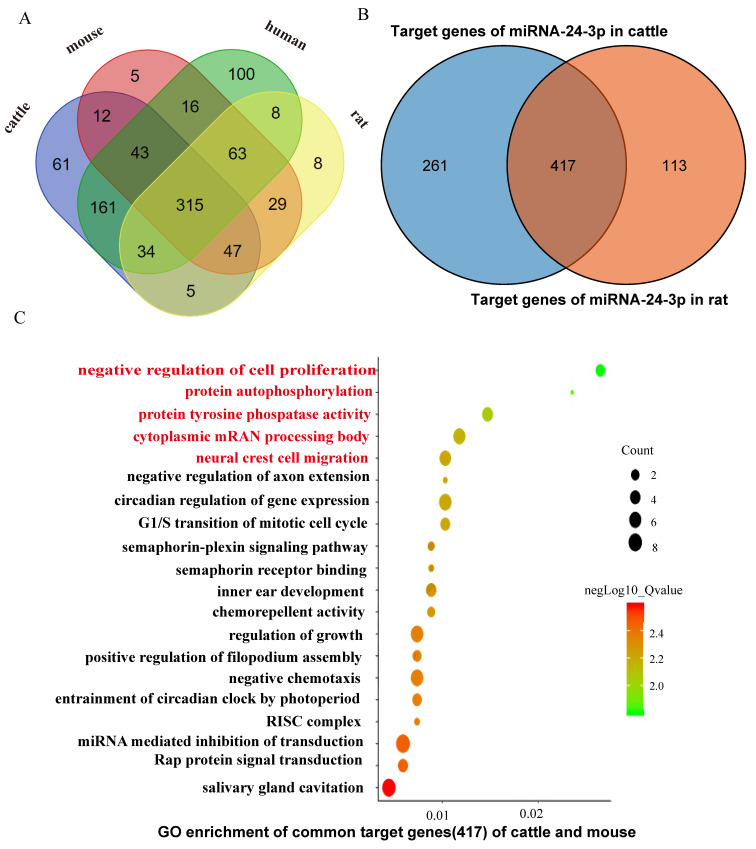
Prediction of miR-24-3p target genes in different species. (**A**) Venn analysis of the target genes of miR-24-3p in cattle, mice, and humans, and rat. (**B**) Venn analysis of the target genes of miR-24-3p in cattle and rat species. (**C**) GO analysis of the common target genes of rat mmu-miR-24-3p and cattle bta-miR-24-3p.

**Figure 5 animals-12-00505-f005:**
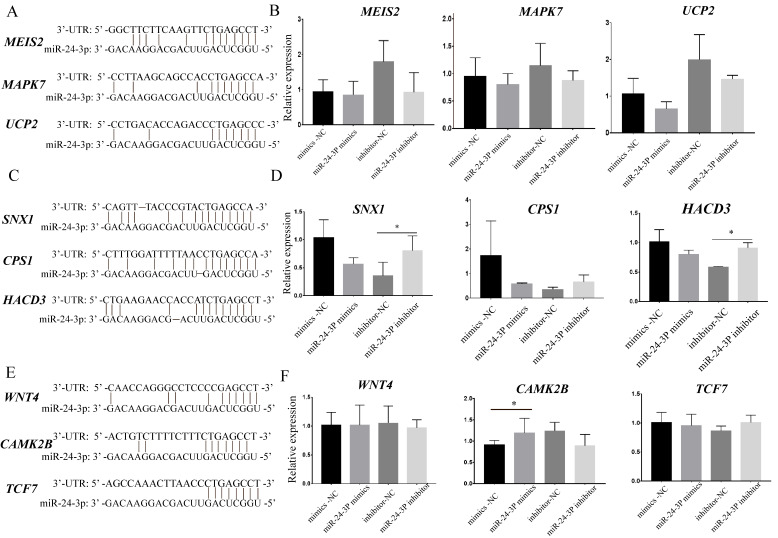
Target gene prediction and their expression profile in miR-24-3p using RT-qPCR in rat and cattle. (**A**,**B**) The target genes shared by mmu-miR-24-3p in cattle and rat were verified by RT-qPCR. (**C**,**D**) The specific target genes of mmu-miR-24-3p related to muscle development were verified by RT-qPCR. (**E**,**F**) The target genes shared by bta-miR-24-3p in cattle and rat were verified by RT-qPCR. mmu-miR-24-3p indicates rat miR-24-3p; bta-miR-24-3p indicates cattle miR-24-3p. * *p* < 0.05.

**Table 1 animals-12-00505-t001:** KEGG analysis of the common target genes of mmu-miR-24-3p and bta-miR-24-3p.

Term	Genes
MAPK signaling pathway	*IL1R1*, *FGFR3*, *RASGRF2*, *TAOK1*, *NLK*, *MAPK14*, *DUSP16*, *PDGFRA*, *RAP1A*, *RAP1B*, *MAPK7*, *DUSP8*, *RASA1*
cAMP signaling pathway	*ATP2B1*, *ATP1B2*, *SSTR1*, *ADORA2A*, *ADCYAP1R1*, *RAP1A*, *PDE3A*, *RAP1B*, *CAMK2B*, *GABBR2*, *RAPGEF3*
Axon guidance	*SEMA5A*, *SEMA6A*, *PLXNA3*, *SEMA4G*, *EPHA8*, *SEMA4B*, *SEMA4A*, *RASA1*
Wnt signaling pathway	*GPC4*, *WNT4*, *TCF7*, *NLK*, *CAMK2B*, *FZD5*, *WNT8B*, *WNT2B*
Basal cell carcinoma	*WNT4*, *TCF7*, *FZD5*, *WNT8B*, *WNT2B*
Signaling pathways regulating pluripotency of stem cells	*ACVR1B*, *WNT4*, *FGFR3*, *MAPK14*, *FZD5*, *WNT8B*, *KAT6A*, *WNT2B*
Pancreatic secretion	*ATP2B1*, *RAB8A*, *ATP2A2*, *ATP1B2*, *RAP1A*, *RAP1B*
Melanogenesis	*WNT4*, *TCF7*, *CAMK2B*, *FZD5*, *WNT8B*, *WNT2B*
Rap1 signaling pathway	*FGFR3*, *MAGI1*, *ADORA2A*, *MAPK14*, *PDGFRA*, *RAP1A*, *RALA*, *RAP1B*, *RAPGEF3*
Hippo signaling pathway	*WNT4*, *TCF7*, *BBC3*, *FZD5*, *PPP2R2D*, *WNT8B*, *WNT2B*
Ras signaling pathway	*FGFR3*, *RAB5B*, *RASGRF2*, *RAB5C*, *PDGFRA*, *RAP1A*, *RALA*, *RAP1B*, *RASA1*

## Data Availability

The muscle transcriptomes of cattle, rat, goat, and pig used in this study were from the NCBI GenBank under accession numbers PRJNA231753, PRJNA718846, PRJNA755813, and PRJNA345427.

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
