# Peer review of "MiR-24-3p Conservatively Regulates Muscle Cell Proliferation and Apoptosis by Targeting Common Gene CAMK2B in Rat and Cattle"

_animals, 2022, doi:10.3390/ani12040505_

Round 1
Reviewer 1 Report
Overall, I think this is an interesting manuscript and increasing understanding of the role of miRNAs in muscle development is important and has application in meat producing livestock species. However, I have a number of concerns such as a lack of detail in the methods and statistical analysis. I also think that much of the results and discussion reported could be improved by removing irrelevant enrichment data that isn't relevant to the species at hand. My specific comments are outlined below.
Line 64 and 65: This sentence is awkwardly worded. I would replace significant with either important or imperative depending on what the author's mean
Line 79: regulated seems to be the wrong form of the word here
Line 83 and 84: I think this should be approved by rather than performed by
Line 90-95: Were these from a single fetus or if not, what stage or age of gestation was the fetus as that can impact where in muscle development the cells are. It is imperative that samples are collected from fetus at similar days of gestation
Line 120 -122 This is unclear
Line 124- the equation is incorrect and should be corrected
Line 125 and 126- insufficient description of analysis
Section 2.7- additional filtering should occur here to remove annotations based on human disease states as there is no relevance in animal production and it confuses the relevant information
Section 2.10- this is not sufficient in its current form- and additional clarification is needed as two sided p-values shouldn't be generated from one-way ANOVA.
Line 175-177: this speculation needs further explanation
Figure 1- resolution is very poor and this cannot be read
Figure 2- as above- resolution is poor
Line 212-214: this is repetitive and poorly worded
Figure 3- as above
Line 234-Line 239- This doesn't make sense as written and needs clarification
Figure 4- as above
Table 1-3- Much of the Terms described have no relevance in animals systems and need to be removed or text needs to be added limiting the interpretation as the annotations are not relevant to the system described
In general there is some redundancy in the results section that should be cleaned up and clarified.
The discussion is interesting but much of it is overstated. The discussion needs to be clarified and material not immediately related to the research questions needs to be removed
Author Response
Dear reviewer,
Thank you for your valuable comments. We have studied the valuable comments from you, the editors carefully, made a significant effort to make the work clearer, and tried our best to revise the manuscript. We highlighted the changes in the revision. The point to point responds to the reviewer’s comments as following:
Comments and Suggestions for Authors
Overall, I think this is an interesting manuscript and increasing understanding of the role of miRNAs in muscle development is important and has application in meat producing livestock species. However, I have a number of concerns such as a lack of detail in the methods and statistical analysis. I also think that much of the results and discussion reported could be improved by removing irrelevant enrichment data that isn't relevant to the species at hand. My specific comments are outlined below.
Line 64 and 65: This sentence is awkwardly worded. I would replace significant with either important or imperative depending on what the author's mean
Response: Thank your valuable comments. We revised the sentence as “Hence, it is important to clarify the molecular mechanism of skeletal muscle development.” in the revision at Line 69-70.
Line 79: regulated seems to be the wrong form of the word here
Response: Thank your valuable comments. We changed the word “regulated” to “regulate” in the revision at Line 83.
Line 83 and 84: I think this should be approved by rather than performed by
Response: Thank your valuable comments. We changed the word “performed” to “approved” in the revision at Line 87.
Line 90-95: Were these from a single fetus or if not, what stage or age of gestation was the fetus as that can impact where in muscle development the cells are. It is imperative that samples are collected from fetus at similar days of gestation
Response: Thank your valuable comments. We revised the sentence in the revision at Line 127 as followings:
“The bovine muscle cell culture samples were aseptically processed for cell culture use”
Line 120 -122 This is unclear
Response: Thank your valuable comments. We revised the sentence in the revision as followings at Line 141-146:
“2.6 Validation of miR-24-3p transfection efficiency in C2C12 cells and primary myoblasts cells
The two groups (miR-24-3p mimics and mimics–negative control, and inhibitor and inhibitor–negative control) were transfected into C2C12 cells and primary my-oblasts cells, which were in the logarithmic phase. After 24h, total RNA was extracted and specifically reversed, and the overexpression efficiency and inhibition efficiency of the miRNA were detected by RT-qPCR as above (Table S2-5).”
Line 124- the equation is incorrect and should be corrected
Response: Thank your valuable comments. We revised the equation “2−△△Ct method” at Line 120 in the revision.
Line 125 and 126- insufficient description of analysis
Response: Thank your valuable comments. We added the description of analysis in the revision as followings at Line 103-120:
“2.3. RNA isolation and reverse transcription-quantitative polymerase chain reaction (RT-qPCR)
The cells were lysed, and total RNA was extracted using TRIzol reagent (Takara) according to the manufacturer’s protocol and transcribed into cDNA using a Reverse Transcription Kit (Takara) [27]. The expression patterns of the target genes and the transcriptional responses of the target genes to the muscle were investigated using re-verse transcription-quantitative polymerase chain reaction (RT-qPCR). According to the manufacturer’s instructions, after DNase treatment, 1000 ng of total RNA was re-verse-transcribed to single-strand cDNA using a HiScript® III 1st Strand cDNA Syn-thesis Kit (+gDNA wiper). The primer pairs of the target genes were used (Table S2-5). Before RT-qPCR analysis, the standard curves for the primer pair of the target genes were generated by regression of the Cq values and a series of ten-fold cDNA dilutions. Primer amplification efficiency was calculated from the slope of the corresponding standard curve, and the efficiency of the target genes. The hypoxic-stable reference gene β-actin was used as the control (Table S2-5). RT-qPCR was performed using the ChamQTM Universal SYBR® qPCR Master Mix (GeneBio Systems, Canada) with the following thermal cycling conditions: 95 â—¦C for 30 s, 40 cycles of 95 â—¦C for 10 s, and 60 â—¦C for 30 s. Each experiment was performed independently three times. The relative expression levels of the target genes were normalized to that of β-actin quantification using the 2−△△Ct method.”
Section 2.7- additional filtering should occur here to remove annotations based on human disease states as there is no relevance in animal production and it confuses the relevant information
Response: Thank your valuable comments. We remove annotations based on human disease states in the revision.
Section 2.10- this is not sufficient in its current form- and additional clarification is needed as two sided p-values shouldn't be generated from one-way ANOVA.
Response: Thank your valuable comments. We revised the sentence in the revision as followings at Line 188-194:
“ 2.11. Statistical analysis
The mean difference between groups was calculated by one-way ANOVA, and the mean difference between the two groups was tested by an independent samples t test. The test data were analyzed using one-way ANOVA with SPSS 23.0 (SPSS Inc., Chica-go, IL, USA) and were expressed as mean ± SE [36]. The level of statistical significance was set at p < 0.05 for all the analyses. All the experiments were performed independently three times (Table S6-9).”
Line 175-177: this speculation needs further explanation
Response: Thank your valuable comments. We revised the sentence as followings at Line 206-208:
“These results showed that miR-24-3p may be an epigenetic regulatory molecule that is not restricted to species and has a broad regulatory role in muscle development (Figure 1, Table S1).”
Figure 1- resolution is very poor and this cannot be read
Response: Thank your valuable comments. We redo and improved the figure quality in the revision.
Figure 2- as above- resolution is poor
Response: Thank your valuable comments. We redo and improved the figure quality in the revision.
Line 212-214: this is repetitive and poorly worded
Response: Thank your valuable comments. We revised the sentence in the revision at Line 246-249.
“The results of the RT-qPCR of miRNA-24-3P in seven tissues (heart, liver, spleen, lung, kidney, muscle, and small intestine) showed that miR-24-3p was highly expressed in muscles and the heart, with these two tissues belonging to skeletal muscle and cardiac muscle, respectively (Figure 3A).”
Figure 3- as above
Response: Thank your valuable comments. We redo and improved the figure quality in the revision.
Line 234-Line 239- This doesn't make sense as written and needs clarification
Response: Thank your valuable comments. We revised the sentence in the revision as followings at Line 280-282:
“There were 315 target genes of miR-24-3p shared among the four species (cattle, mice, humans, and rats), accounting for 46.46% of the total number of miR-24-3p target genes according to the Venn analysis”
Figure 4- as above
Response: Thank your valuable comments. We redo and improved the figure quality in the revision.
Table 1-3- Much of the Terms described have no relevance in animals systems and need to be removed or text needs to be added limiting the interpretation as the annotations are not relevant to the system described
Response: Thank your valuable comments. We removed or text needs to be added limiting the interpretation as the annotations are not relevant to the system described
In general there is some redundancy in the results section that should be cleaned up and clarified.
Response: Thank your valuable comments.
We revised the figure order of section “ 3.2. MiR-24-3p promotes C2C12 cells proliferation and inhibits its apoptosis”.
We revised the Figure 4 and removed the Table 2-3 and we removed the description of Table 2-3 in the section “ 3.4. Prediction of miR-24-3p target genes and annotation in different species”
We also rewrote the section 3.5 as followings:
“3.5. The specific and common target genes' expression profile of miR-24-3p in cattle and rats
The double luciferase test was performed for miR-24-3p and their predicted target genes using 293T cells that were transfected with miR-24-3p mimics / mimics NC / inhibitor / inhibitor-NC (Figure 5A, C, and E). The unique target genes MEIS2, MAPK7, and UCP2 in rats were targeted by mmu-miR-24-3p (Figure 5A). The unique target genes SNX1, HACD3, and CPS1 in cattle were targeted by bta-miR-24-3p (Figure 5C; Table S6-7). The results showed that the shared target genes WNT4, CAMK2B, and TCF7 were targeted by mmu-miR-24-3p and bta-miR-24-3p (Figure 5E). For cattle, after inhibition of bta-miR-24-3p, the expression of the SNX1 and HACD3 genes increased significantly (Figure 5D). Of these three common target genes (WNT4, CAMK2B, and TCF7), we found that after overexpression of miR-24-3p, the expression of the tar-get gene CAMK2B increased significantly (Figure 5F, Table S8-9).”
The discussion is interesting but much of it is overstated. The discussion needs to be clarified and material not immediately related to the research questions needs to be removed
Response: Thank your valuable comments. We revised the discussion section as followings at Line 367-408:
“In order to verify the function of miR-24-3p in rat and cattle muscle development, this experiment used C2C12 cells and bovine muscle cells to verify the effect of miR-24-3p on rat and cattle muscle cell proliferation and apoptosis. We detected the proliferation-related genes' expression level, such as CYCLINE [51], PCNA [52], BAX [53], CASPASE9, CASPASE3, CASPASE8 [54] in rats, BCL2 [55], CDK2 [56], and CASPASE9 in cattle [54]. In rats, the C2C12 cell number was significantly increased with miR-24-3p mimics compared to the mimics–negative control (mimics-NC) group. After miR-24-3p was overexpressed, the proliferation-related genes (CYCLINE, PCNA, and BAX) were significantly increased; however, the apoptosis-related genes (CASPASE9, CASPASE3, and CASPASE8) were significantly decreased. These three results show that miR-24-3p could significantly promote C2C12 cell proliferation and inhibit its apoptosis (Figure 2) [57]. In cattle, the RT-qPCR results for miR-24-3p in seven different tissues showed that miR-24-3p was highly expressed in muscle (skeletal muscle) and the heart (cardiac muscle) compared to the other five tissues, indicating that miR-24-3p plays an important role in muscle development. Furthermore, the OD value of bovine muscle cells in the miR-24-3p group was significantly increased compared to the NC group. The expression level of proliferation-related genes (BCL2 and CDK2) were significantly increased after miR-24-3p was overexpressed, while the expression of proliferation key gene (CASPASE9) genes was significantly decreased after miR-24-3p was inhibited. These four results showed that miR-24-3p can promote bovine muscle cell proliferation and inhibits their apoptosis (Figure 3) [57]. This study found, for the first time, that miR-24-3p also has functions related to the regulation of muscle cell development in bovine muscle cells. This is consistent with the existing re-ports on the regulatory effect of miR-24-3p on muscle development in mice and humans [57].
The GO and KEGG analysis of the target genes shared by mmu-miR-24-3p and bta-miR-24-3p showed enrichment of important cell functions and signal pathways related to muscle development, including negative regulation of cell proliferation, protein autophosphorylation, the MAPK signaling pathway, the cAMP signaling pathway, Axon guidance, the Wnt signaling pathway, and basal cell carcinoma. These results show that the target genes shared by mmu-miR-24-3p and bta-miR-24-3p are enriched in multiple cell functions and signal pathways that are closely related to muscle development (Figure 4). The shared target genes WNT4, CAMK2B, and TCF7 were targeted by mmu-miR-24-3p and bta-miR-24-3p, which was proved by the double luciferase test performed for miR-24-3p and their predicted target genes using 293T cells that were transfected with miR-24-3p mimics / mimics NC / inhibitor / inhibitor-NC. After overexpression of miR-24-3p, the expression of the shared target gene CAMK2B increased significantly. Previous studies showed that CAMK2B play an important role in muscle development [58-59]. These results show that CAMK2B is a conserved target gene of miR-24-3p in cattle and rats that plays an important role in muscle development in cattle and rats, which is consistent with the results of previous studies (Figure 5) [58-59].”
Thank you for your consideration our manuscript of “Conserved miR-24-3p regulates muscle cell proliferation and apoptosis in rats and cattle " publish on Animals.
We are looking forward to hearing from you.
Yours sincerely,
Ruihua Dang

Reviewer 2 Report
Interesting and novel paper. My suggestions:
1) Language editing (or more careful proof-reading) is highly recommended. There are several sentences that don't have an end, such as in line 13-14.
2) I would recommend writing out every acronym when first mentioned, including names of genes (e.g., Grh2, SOX7, etc.) Some definitions would also be helpful (e.g., C2C12 cells, Dicer, etc.). I would suggest also defining mmu, hsa, bta when first mentioned in text or figure/table.
3) The figures in the manuscript are of very bad quality and not legible at all. Please replace them with higher quality graphs or provide direct link to each figure (if the paper will be published online only).
Author Response
Dear reviewer,
Thank you for your valuable comments. We have studied the valuable comments from you, the editors carefully, made a significant effort to make the work clearer, and tried our best to revise the manuscript. We highlighted the changes in the revision. The point to point responds to the reviewer’s comments as following:
Comments and Suggestions for Authors
Interesting and novel paper. My suggestions:
1) Language editing (or more careful proof-reading) is highly recommended. There are several sentences that don't have an end, such as in line 13-14.
Response: Thank your valuable comments. We submit our manuscript to MDPI to improve English language.
We revised the sentence in the revision at line 13-14 as followings:
“ There is a large proportion of the target genes of miR-24-3p shared by cattle, rats, goats, and pigs.”
2) I would recommend writing out every acronym when first mentioned, including names of genes (e.g., Grh2, SOX7, etc.) Some definitions would also be helpful (e.g., C2C12 cells, Dicer, etc.). I would suggest also defining mmu, hsa, bta when first mentioned in text or figure/table.
Response: Thank your valuable comments. We added every acronym in the revision as followings:
“GRB2 (growth factor receptor-bound protein 2) at Line 55;
IGFBP5 (insulin-like growth factor binding protein 5) at Line 59;
SOX7 (SRY-box transcription factor 7) at Line 63;
MEIS2 (Meis Homeobox 2), MAPK7 (Mitogen-Activated Protein Kinase 7), UCP2 (Uncoupling Protein 2), SNX1 (Sorting Nexin 1), CPS1 (Carbamoyl-Phosphate Synthase 1), HACD3 (3-Hydroxyacyl-CoA Dehydratase 3), WNT4 (Wnt Family Member 4), CAMK2B (Calcium/Calmodulin Dependent Protein Kinase II Beta), TCF7 (Transcription Factor 7) at Line 164-168;
We added the note “mmu-miR-24-3p indicates rat miR-24-3p; bta-miR-24-3p indicates cattle miR-24-3p” at Line 333-334.”
3) The figures in the manuscript are of very bad quality and not legible at all. Please replace them with higher quality graphs or provide direct link to each figure (if the paper will be published online only).
Response: Thank your valuable comments. We redo our figures to make it clear and higher quality.
We also revised our methods section as followings:
“2.3. RNA isolation and reverse transcription-quantitative polymerase chain reaction (RT-qPCR)
The cells were lysed, and total RNA was extracted using TRIzol reagent (Takara) according to the manufacturer’s protocol and transcribed into cDNA using a Reverse Transcription Kit (Takara) [27]. The expression patterns of the target genes and the transcriptional responses of the target genes to the muscle were investigated using re-verse transcription-quantitative polymerase chain reaction (RT-qPCR). According to the manufacturer’s instructions, after DNase treatment, 1000 ng of total RNA was re-verse-transcribed to single-strand cDNA using a HiScript® III 1st Strand cDNA Syn-thesis Kit (+gDNA wiper). The primer pairs of the target genes were used (Table S2-5). Before RT-qPCR analysis, the standard curves for the primer pair of the target genes were generated by regression of the Cq values and a series of ten-fold cDNA dilutions. Primer amplification efficiency was calculated from the slope of the corresponding standard curve, and the efficiency of the target genes. The hypoxic-stable reference gene β-actin was used as the control (Table S2-5). RT-qPCR was performed using the ChamQTM Universal SYBR® qPCR Master Mix (GeneBio Systems, Canada) with the following thermal cycling conditions: 95 â—¦C for 30 s, 40 cycles of 95 â—¦C for 10 s, and 60 â—¦C for 30 s. Each experiment was performed independently three times. The relative expression levels of the target genes were normalized to that of β-actin quantification using the 2−△△Ct method.
2.4. Collection of cell samples (C2C12, 239T, and Qinchuan cattle fetus)
The C2C12 cell line (accession number: CRL-1772) and the 293T cell line (acces-sion number: CRL-3216) were obtained from Procell Life Science & Technology Co., Ltd. [28,29]. The 293T cell line (accession number: NM_131629) was obtained from Procell Life Science & Technology Co., Ltd. The Qinchuan cattle fetus (3 months old) samples were collected from a beef cattle slaughterhouse near Xi'an. The tissue spectrum in-cluded the heart, liver, spleen, lung, kidney, muscle, and small intestine. The bovine muscle cell culture samples were aseptically processed for cell culture use. The study was approved by the Biomedical Ethics Committee of Northwest A&F University.
2.5. Cell culture
The C2C12 cells were procured from Procell Life Science & Technology Co., Ltd. The 293T cells were procured from Procell Life Science & Technology Co., Ltd. These C2C12 cells, the bovine muscle cells, and the 293T cells were maintained at 37℃in a 5% CO2 humidified incubator and were grown in Dulbecco’s Modified Eagle’s Medium (DEME) supplemented with 10% fetal bovine serum (FBS) [30]. When the confluence of cells reached 80-90%, they were digested with 0.25% trypsin and passaged to a new culture dish at a ratio of 1:3. The cells were transfected with 50 nM of control or mimic for miR-24b-3p mixed with Opti-MEM and Lipofectamine RNAiMAX (Invitrogen) ac-cording to manufacturer’s protocol [31]. All the analyses were performed in triplicate.
2.6 Validation of miR-24-3p transfection efficiency in C2C12 cells and primary myoblasts cells
The two groups (miR-24-3p mimics and mimics–negative control, and inhibitor and inhibitor–negative control) were transfected into C2C12 cells and primary my-oblasts cells, which were in the logarithmic phase. After 24h, total RNA was extracted and specifically reversed, and the overexpression efficiency and inhibition efficiency of the miRNA were detected by RT-qPCR as above (Table S2-5).
2.11. Statistical analysis
The mean difference between groups was calculated by one-way ANOVA, and the mean difference between the two groups was tested by an independent samples t test. The test data were analyzed using one-way ANOVA with SPSS 23.0 (SPSS Inc., Chica-go, IL, USA) and were expressed as mean ± SE [36]. The level of statistical significance was set at p < 0.05 for all the analyses. All the experiments were performed independently three times (Table S6-9).”
We also revised our results section as followings:
“3.4. Prediction of miR-24-3p target genes and annotation in different species
There were 315 target genes of miR-24-3p shared among the four species (cattle, mice, humans, and rats), accounting for 46.46% of the total number of miR-24-3p tar-get genes according to the Venn analysis. This may be an important basis by which miR-24-3p conservatively regulates muscle development in different species (Figure 4A, Table 1). There were 417 target genes shared between the two species (rat and cattle), and these common target genes account for 78.7% of all the target genes in rats and 61.5% of all the target genes in cattle (Figure 4B, Table 1-3). The large proportion of the same target genes may be related to the previously demonstrated ability of miR-24-3p to regulate the proliferation and apoptosis of C2C12 cell lines and bovine primary muscle cells (Figure 4, Table 1-3).
3.5. The specific and common target genes' expression profile of miR-24-3p in cattle and rats
The double luciferase test was performed for miR-24-3p and their predicted target genes using 293T cells that were transfected with miR-24-3p mimics / mimics NC / in-hibitor / inhibitor-NC (Figure 5A, C, and E). The unique target genes MEIS2, MAPK7, and UCP2 in rats were targeted by mmu-miR-24-3p (Figure 5A). The unique target genes SNX1, HACD3, and CPS1 in cattle were targeted by bta-miR-24-3p (Figure 5C; Table S6-7). The results showed that the shared target genes WNT4, CAMK2B, and TCF7 were targeted by mmu-miR-24-3p and bta-miR-24-3p (Figure 5E). For cattle, after inhibition of bta-miR-24-3p, the expression of the SNX1 and HACD3 genes increased significantly (Figure 5D). Of these three common target genes (WNT4, CAMK2B, and TCF7), we found that after overexpression of miR-24-3p, the expression of the target gene CAMK2B increased significantly (Figure 5F, Table S8-9).”
We also revised our discussion section as followings at Line 352-391:
“In order to verify the function of miR-24-3p in rat and cattle muscle development, this experiment used C2C12 cells and bovine muscle cells to verify the effect of miR-24-3p on rat and cattle muscle cell proliferation and apoptosis. We detected the proliferation-related genes' expression level, such as CYCLINE [51], PCNA [52], BAX [53], CASPASE9, CASPASE3, CASPASE8 [54] in rats, BCL2 [55], CDK2 [56], and CASPASE9 in cattle [54]. In rats, the C2C12 cell number was significantly increased with miR-24-3p mimics compared to the mimics–negative control (mimics-NC) group. After miR-24-3p was overexpressed, the proliferation-related genes (CYCLINE, PCNA, and BAX) were significantly increased; however, the apoptosis-related genes (CASPASE9, CASPASE3, and CASPASE8) were significantly decreased. These three results show that miR-24-3p could significantly promote C2C12 cell proliferation and inhibit its apoptosis (Figure 2) [57]. In cattle, the RT-qPCR results for miR-24-3p in seven different tissues showed that miR-24-3p was highly expressed in muscle (skeletal muscle) and the heart (cardiac muscle) compared to the other five tissues, indicating that miR-24-3p plays an important role in muscle development. Furthermore, the OD value of bovine muscle cells in the miR-24-3p group was significantly increased compared to the NC group. The expression level of proliferation-related genes (BCL2 and CDK2) were significantly increased after miR-24-3p was overexpressed, while the expression of proliferation key gene (CASPASE9) genes was significantly decreased after miR-24-3p was inhibited. These four results showed that miR-24-3p can promote bovine muscle cell proliferation and inhibits their apoptosis (Figure 3) [57]. This study found, for the first time, that miR-24-3p also has functions related to the regulation of muscle cell development in bovine muscle cells. This is consistent with the existing re-ports on the regulatory effect of miR-24-3p on muscle development in mice and humans [57].
The GO and KEGG analysis of the target genes shared by mmu-miR-24-3p and bta-miR-24-3p showed enrichment of important cell functions and signal pathways related to muscle development, including negative regulation of cell proliferation, protein autophosphorylation, the MAPK signaling pathway, the cAMP signaling pathway, Axon guidance, the Wnt signaling pathway, and basal cell carcinoma. These results show that the target genes shared by mmu-miR-24-3p and bta-miR-24-3p are enriched in multiple cell functions and signal pathways that are closely related to muscle development (Figure 4). The shared target genes WNT4, CAMK2B, and TCF7 were targeted by mmu-miR-24-3p and bta-miR-24-3p, which was proved by the double luciferase test performed for miR-24-3p and their predicted target genes using 293T cells that were transfected with miR-24-3p mimics / mimics NC / inhibitor / inhibitor-NC. After overexpression of miR-24-3p, the expression of the shared target gene CAMK2B increased significantly. Previous studies showed that CAMK2B play an important role in muscle development [58-59]. These results show that CAMK2B is a conserved target gene of miR-24-3p in cattle and rats that plays an important role in muscle development in cattle and rats, which is consistent with the results of previous studies (Figure 5) [58-59].”
We also revised our conclusion section as followings at Line 352-391:
“In summary, with the miRNA transcriptome sequencing data of muscle tissue in cattle, rats, goats, and pigs, we found that miR-24-3p plays a potential role in regulating muscle development in animals. CCK-8 and RT-qPCR analysis showed that mmu-miR-24-3p can positively regulate C2C12 cell proliferation and apoptosis, and bta-miR-24-3p can also positively regulate the proliferation and apoptosis of bovine muscle primary cells. The GO and KEGG enrichment analysis results showed that the target genes of miR-24-3p in cattle, rats, goats, and pigs are closely related to muscle development. The shared target genes WNT4, CAMK2B, and TCF7 were targeted by both mmu-miR-24-3p and bta-miR-24-3p using the double luciferase test. After inhibition of miR-24-3p, the target gene CAMK2B, which plays an important role in muscle development, increased significantly, indicating that miR-24-3p is a conservative miRNA and that it can regulate the most predicted target gene (CAMK2B) to influence the development of muscle.”
Thank you for your consideration our manuscript of “Conserved miR-24-3p regulates muscle cell proliferation and apoptosis in rats and cattle " publish on Animals.
We are looking forward to hearing from you.
Yours sincerely,
Ruihua Dang

Reviewer 3 Report
In this study, the author compared the miRNA transcriptome sequencing data of cattle, rats, goats and pigs muscle tissue using the NCBI database. They found a conserved miR-24-3p that may regulate muscle development in the four species.
Overexpression and inhibition of miR-24-3p in C2C12 (mmu-miR-24-3p) and bovine primary myoblasts cell lines (bta-miR-24-3p) confirmed the suggestion that this mi-RNA positively regulates cell proliferation and apoptosis by regulating key proliferation and apoptosis genes in muscle development.
The work have big problems. All figures were of such poor quality that one could only guess the results. The second problem was the poor quality of the english language, which made the article reading very difficult and confusing.
It is not clear, why the author haven´t chosen exactly the same key genes for proliferation and apoptosis in the overexpression and inhibition experiments for mouse and cattle. In addition, the gene targets studied using mmu-miR-24-3p and bta-miR-24-3p were very different, so that a comparison of the results is confusing. The explanation and the meaning for the resulted target genes in muscle proliferation and apoptosis were missing in the discussion.
I cannot accept the article.
Author Response
Dear reviewer,
Thank you for your valuable comments. We have studied the valuable comments from you, the editors carefully, made a significant effort to make the work clearer, and tried our best to revise the manuscript. We highlighted the changes in the revision. The point to point responds to the reviewer’s comments as following:
Comments and Suggestions for Authors
In this study, the author compared the miRNA transcriptome sequencing data of cattle, rats, goats and pigs muscle tissue using the NCBI database. They found a conserved miR-24-3p that may regulate muscle development in the four species.
Overexpression and inhibition of miR-24-3p in C2C12 (mmu-miR-24-3p) and bovine primary myoblasts cell lines (bta-miR-24-3p) confirmed the suggestion that this mi-RNA positively regulates cell proliferation and apoptosis by regulating key proliferation and apoptosis genes in muscle development.
The work have big problems. All figures were of such poor quality that one could only guess the results. The second problem was the poor quality of the english language, which made the article reading very difficult and confusing.
Response: Thank your valuable comments. We redo our figures to make it clear and higher quality in the revision. We submit our manuscript to MDPI to improve English language.
It is not clear, why the author haven´t chosen exactly the same key genes for proliferation and apoptosis in the overexpression and inhibition experiments for mouse and cattle. In addition, the gene targets studied using mmu-miR-24-3p and bta-miR-24-3p were very different, so that a comparison of the results is confusing. The explanation and the meaning for the resulted target genes in muscle proliferation and apoptosis were missing in the discussion.
Response: Thank your valuable comments. We redo our figures to make it clear and higher quality. In this study, we found that the shared target genes WNT4, CAMK2B and TCF7 were targeted by both mmu-miR-24-3p and bta-miR-24-3p using double luciferase test. Of three common target genes, we found that after overexpression of miR-24-3p, the expression of the target gene CAMK2B increased significantly (Figure 5F, Table S8-9).
We also added the discussion of these shared target gene CAMK2B as followings at Line 399-408:
“The shared target genes WNT4, CAMK2B, and TCF7 were targeted by mmu-miR-24-3p and bta-miR-24-3p, which was proved by the double luciferase test performed for miR-24-3p and their predicted target genes using 293T cells that were transfected with miR-24-3p mimics / mimics NC / inhibitor / inhibitor-NC. After overexpression of miR-24-3p, the expression of the shared target gene CAMK2B increased significantly. Previous studies showed that CAMK2B plays an important role in muscle development [58-59]. These results show that CAMK2B is a conserved target gene of miR-24-3p in cattle and rats that plays an important role in muscle development in cattle and rats, which is consistent with the results of previous studies (Figure 5) [58-59].”
We also revised our methods section as followings:
“2.3. RNA isolation and reverse transcription-quantitative polymerase chain reaction (RT-qPCR)
The cells were lysed, and total RNA was extracted using TRIzol reagent (Takara) according to the manufacturer’s protocol and transcribed into cDNA using a Reverse Transcription Kit (Takara) [27]. The expression patterns of the target genes and the transcriptional responses of the target genes to the muscle were investigated using re-verse transcription-quantitative polymerase chain reaction (RT-qPCR). According to the manufacturer’s instructions, after DNase treatment, 1000 ng of total RNA was re-verse-transcribed to single-strand cDNA using a HiScript® III 1st Strand cDNA Syn-thesis Kit (+gDNA wiper). The primer pairs of the target genes were used (Table S2-5). Before RT-qPCR analysis, the standard curves for the primer pair of the target genes were generated by regression of the Cq values and a series of ten-fold cDNA dilutions. Primer amplification efficiency was calculated from the slope of the corresponding standard curve, and the efficiency of the target genes. The hypoxic-stable reference gene β-actin was used as the control (Table S2-5). RT-qPCR was performed using the ChamQTM Universal SYBR® qPCR Master Mix (GeneBio Systems, Canada) with the following thermal cycling conditions: 95 â—¦C for 30 s, 40 cycles of 95 â—¦C for 10 s, and 60 â—¦C for 30 s. Each experiment was performed independently three times. The relative expression levels of the target genes were normalized to that of β-actin quantification using the 2−△△Ct method.
2.4. Collection of cell samples (C2C12, 239T, and Qinchuan cattle fetus)
The C2C12 cell line (accession number: CRL-1772) and the 293T cell line (accession number: CRL-3216) were obtained from Procell Life Science & Technology Co., Ltd. [28,29]. The 293T cell line (accession number: NM_131629) was obtained from Procell Life Science & Technology Co., Ltd. The Qinchuan cattle fetus (3 months old) samples were collected from a beef cattle slaughterhouse near Xi'an. The tissue spectrum included the heart, liver, spleen, lung, kidney, muscle, and small intestine. The bovine muscle cell culture samples were aseptically processed for cell culture use. The study was approved by the Biomedical Ethics Committee of Northwest A&F University.
2.5. Cell culture
The C2C12 cells were procured from Procell Life Science & Technology Co., Ltd. The 293T cells were procured from Procell Life Science & Technology Co., Ltd. These C2C12 cells, the bovine muscle cells, and the 293T cells were maintained at 37℃in a 5% CO2 humidified incubator and were grown in Dulbecco’s Modified Eagle’s Medium (DEME) supplemented with 10% fetal bovine serum (FBS) [30]. When the confluence of cells reached 80-90%, they were digested with 0.25% trypsin and passaged to a new culture dish at a ratio of 1:3. The cells were transfected with 50 nM of control or mimic for miR-24b-3p mixed with Opti-MEM and Lipofectamine RNAiMAX (Invitrogen) ac-cording to manufacturer’s protocol [31]. All the analyses were performed in triplicate.
2.6 Validation of miR-24-3p transfection efficiency in C2C12 cells and primary myoblasts cells
The two groups (miR-24-3p mimics and mimics–negative control, and inhibitor and inhibitor–negative control) were transfected into C2C12 cells and primary my-oblasts cells, which were in the logarithmic phase. After 24h, total RNA was extracted and specifically reversed, and the overexpression efficiency and inhibition efficiency of the miRNA were detected by RT-qPCR as above (Table S2-5).
2.11. Statistical analysis
The mean difference between groups was calculated by one-way ANOVA, and the mean difference between the two groups was tested by an independent samples t test. The test data were analyzed using one-way ANOVA with SPSS 23.0 (SPSS Inc., Chica-go, IL, USA) and were expressed as mean ± SE [36]. The level of statistical significance was set at p < 0.05 for all the analyses. All the experiments were performed independently three times (Table S6-9).”
We also revised our results section as followings:
“3.4. Prediction of miR-24-3p target genes and annotation in different species
There were 315 target genes of miR-24-3p shared among the four species (cattle, mice, humans, and rats), accounting for 46.46% of the total number of miR-24-3p tar-get genes according to the Venn analysis. This may be an important basis by which miR-24-3p conservatively regulates muscle development in different species (Figure 4A, Table 1). There were 417 target genes shared between the two species (rat and cattle), and these common target genes account for 78.7% of all the target genes in rats and 61.5% of all the target genes in cattle (Figure 4B, Table 1-3). The large proportion of the same target genes may be related to the previously demonstrated ability of miR-24-3p to regulate the proliferation and apoptosis of C2C12 cell lines and bovine primary muscle cells (Figure 4, Table 1-3).
3.5. The specific and common target genes' expression profile of miR-24-3p in cattle and rats
The double luciferase test was performed for miR-24-3p and their predicted target genes using 293T cells that were transfected with miR-24-3p mimics / mimics NC / in-hibitor / inhibitor-NC (Figure 5A, C, and E). The unique target genes MEIS2, MAPK7, and UCP2 in rats were targeted by mmu-miR-24-3p (Figure 5A). The unique target genes SNX1, HACD3, and CPS1 in cattle were targeted by bta-miR-24-3p (Figure 5C; Table S6-7). The results showed that the shared target genes WNT4, CAMK2B, and TCF7 were targeted by mmu-miR-24-3p and bta-miR-24-3p (Figure 5E). For cattle, after inhibition of bta-miR-24-3p, the expression of the SNX1 and HACD3 genes increased significantly (Figure 5D). Of these three common target genes (WNT4, CAMK2B, and TCF7), we found that after overexpression of miR-24-3p, the expression of the target gene CAMK2B increased significantly (Figure 5F, Table S8-9).”
We also revised our discussion section as followings at Line 352-391:
“In order to verify the function of miR-24-3p in rat and cattle muscle development, this experiment used C2C12 cells and bovine muscle cells to verify the effect of miR-24-3p on rat and cattle muscle cell proliferation and apoptosis. We detected the proliferation-related genes' expression level, such as CYCLINE [51], PCNA [52], BAX [53], CASPASE9, CASPASE3, CASPASE8 [54] in rats, BCL2 [55], CDK2 [56], and CASPASE9 in cattle [54]. In rats, the C2C12 cell number was significantly increased with miR-24-3p mimics compared to the mimics–negative control (mimics-NC) group. After miR-24-3p was overexpressed, the proliferation-related genes (CYCLINE, PCNA, and BAX) were significantly increased; however, the apoptosis-related genes (CASPASE9, CASPASE3, and CASPASE8) were significantly decreased. These three results show that miR-24-3p could significantly promote C2C12 cell proliferation and inhibit its apoptosis (Figure 2) [57]. In cattle, the RT-qPCR results for miR-24-3p in seven different tissues showed that miR-24-3p was highly expressed in muscle (skeletal muscle) and the heart (cardiac muscle) compared to the other five tissues, indicating that miR-24-3p plays an important role in muscle development. Furthermore, the OD value of bovine muscle cells in the miR-24-3p group was significantly increased compared to the NC group. The expression level of proliferation-related genes (BCL2 and CDK2) were significantly increased after miR-24-3p was overexpressed, while the expression of proliferation key gene (CASPASE9) genes was significantly decreased after miR-24-3p was inhibited. These four results showed that miR-24-3p can promote bovine muscle cell proliferation and inhibits their apoptosis (Figure 3) [57]. This study found, for the first time, that miR-24-3p also has functions related to the regulation of muscle cell development in bovine muscle cells. This is consistent with the existing re-ports on the regulatory effect of miR-24-3p on muscle development in mice and humans [57].
The GO and KEGG analysis of the target genes shared by mmu-miR-24-3p and bta-miR-24-3p showed enrichment of important cell functions and signal pathways related to muscle development, including negative regulation of cell proliferation, protein autophosphorylation, the MAPK signaling pathway, the cAMP signaling pathway, Axon guidance, the Wnt signaling pathway, and basal cell carcinoma. These results show that the target genes shared by mmu-miR-24-3p and bta-miR-24-3p are enriched in multiple cell functions and signal pathways that are closely related to muscle development (Figure 4). The shared target genes WNT4, CAMK2B, and TCF7 were targeted by mmu-miR-24-3p and bta-miR-24-3p, which was proved by the double luciferase test performed for miR-24-3p and their predicted target genes using 293T cells that were transfected with miR-24-3p mimics / mimics NC / inhibitor / inhibitor-NC. After overexpression of miR-24-3p, the expression of the shared target gene CAMK2B increased significantly. Previous studies showed that CAMK2B play an important role in muscle development [58-59]. These results show that CAMK2B is a conserved target gene of miR-24-3p in cattle and rats that plays an important role in muscle development in cattle and rats, which is consistent with the results of previous studies (Figure 5) [58-59].”
We also revised our conclusion section as followings at Line 352-391:
“In summary, with the miRNA transcriptome sequencing data of muscle tissue in cattle, rats, goats, and pigs, we found that miR-24-3p plays a potential role in regulating muscle development in animals. CCK-8 and RT-qPCR analysis showed that mmu-miR-24-3p can positively regulate C2C12 cell proliferation and apoptosis, and bta-miR-24-3p can also positively regulate the proliferation and apoptosis of bovine muscle primary cells. The GO and KEGG enrichment analysis results showed that the target genes of miR-24-3p in cattle, rats, goats, and pigs are closely related to muscle development. The shared target genes WNT4, CAMK2B, and TCF7 were targeted by both mmu-miR-24-3p and bta-miR-24-3p using the double luciferase test. After inhibition of miR-24-3p, the target gene CAMK2B, which plays an important role in muscle development, increased significantly, indicating that miR-24-3p is a conservative miRNA and that it can regulate the most predicted target gene (CAMK2B) to influence the development of muscle.”
I cannot accept the article.
Thank you for your consideration our manuscript of “Conserved miR-24-3p regulates muscle cell proliferation and apoptosis in rats and cattle " publish on Animals.
We are looking forward to hearing from you.
Yours sincerely,
Ruihua Dang

Round 2
Reviewer 1 Report
The manuscript is much improved. A few additional comments are below.
Line 123- 3 months old should be replaced with in third month of gestation
Section 2.3 needs to come after section 2.4 and 2.5 to improve readability - Cells need to be grown before they can be lysed and extracted.
Figure 2 is still difficult to read
Author Response
Dear reviewer,
Thank you for your valuable comments. We have studied the valuable comments from you carefully, made a significant effort to make the work clearer, and tried our best to revise the manuscript. We highlighted the changes in the revision. The point to point responds to the reviewer’s comments as following:
Comments and Suggestions for Authors
The manuscript is much improved. A few additional comments are below.
Line 123- 3 months old should be replaced with in third month of gestation
Response: Thank your valuable comments. We changed “3 months old” to “ in third month of gestation” at Line 105.
Section 2.3 needs to come after section 2.4 and 2.5 to improve readability - Cells need to be grown before they can be lysed and extracted.
Response: Thank your valuable comments. We placed section 2.3 come after section 2.4 and 2.5 in the revision.
Figure 2 is still difficult to read
Response: Thank your valuable comments. We redo and improved the figure 2 quality in the revision.
Thank you for your consideration our manuscript of “Conserved miR-24-3p regulates muscle cell proliferation and apoptosis in rats and cattle " publish on Animals.
We are looking forward to hearing from you.
Yours sincerely,
Ruihua Dang

Reviewer 3 Report
-The images are now of acceptable quality
-The submission of the manuscript to MDPI to improve English language is a good decision.
-The authors changed the manuscript at different paragraphs, so that reading and understanding it has become easier.
Major changes:
WNT4, CAMK2B and TCF7are all targets in the Wnt signalling pathway. Discus the role of Wnt signalling in muscle proliferation.
Minor changes
- 7 (145-148) is Cursive it should be changed. 2.8 (148) must be a new paragraph
- Finger 2 (229) in miR-148a-3p is not correct
- (24-243) the word Lower is missed
- (349 and 358 and 377) the Figures should not be mentioned again and the citation is not in the correct position. In row 362 the citation is correct and enough.
- The citation for the target genes in page 11 (342-343) is also not in the right position.
Author Response
Dear reviewer,
Thank you for your valuable comments. We have studied the valuable comments from you carefully, made a significant effort to make the work clearer, and tried our best to revise the manuscript. We highlighted the changes in the revision. The point to point responds to the reviewer’s comments as following:
Comments and Suggestions for Authors
-The images are now of acceptable quality
-The submission of the manuscript to MDPI to improve English language is a good decision.
-The authors changed the manuscript at different paragraphs, so that reading and understanding it has become easier.
Major changes:
WNT4, CAMK2B and TCF7are all targets in the Wnt signalling pathway. Discus the role of Wnt signalling in muscle proliferation.
Response: Thank your valuable comments. We added the sentences in the discussion section in the revision at Line 374-376 as followings:
“Moreover, the three shared target genes (WNT4, CAMK2B and TCF7) were all targets in the Wnt signaling pathway, which was consistent with the previous studies claimed that Wnt signaling pathway play an important role in muscle proliferation [58-59].”
Minor changes
7 (145-148) is Cursive it should be changed. 2.8 (148) must be a new paragraph
Response: Thank your valuable comments. We changed the section 2.7 and placed 2.8 as a new paragraph.
Finger 2 (229) in miR-148a-3p is not correct
Response: Thank your valuable comments. We changed the word “ miR-148a-3p ” to “ miR-24-3p ” at Line 233, 261.
(24-243) the word Lower is missed
Response: Thank your valuable comments. We added the word “lower” at Line 247.
(349 and 358 and 377) the Figures should not be mentioned again and the citation is not in the correct position. In row 362 the citation is correct and enough.
Response: Thank your valuable comments. We revised the citation in row 363.
The citation for the target genes in page 11 (342-343) is also not in the right position.
Response: Thank your valuable comments. We revised the sentence in the revision at Line 347-349 as followings:
“After miR-24-3p was overexpressed, the expression level of three proliferation-related genes CYCLINE, PCNA, and BAX were significantly increased; however, the expression level of three apoptosis-related genes CASPASE9, CASPASE3, and CASPASE8 were significantly decreased.”
Thank you for your consideration our manuscript of “Conserved miR-24-3p regulates muscle cell proliferation and apoptosis in rats and cattle " publish on Animals.
We are looking forward to hearing from you.
Yours sincerely,
Ruihua Dang
